# Recent Advances and Future Prospects in Immune Checkpoint (ICI)-Based Combination Therapy for Advanced HCC

**DOI:** 10.3390/cancers13081949

**Published:** 2021-04-18

**Authors:** Yawen Dong, Jeffrey Sum Lung Wong, Ryohichi Sugimura, Ka-On Lam, Bryan Li, Gerry Gin Wai Kwok, Roland Leung, Joanne Wing Yan Chiu, Tan To Cheung, Thomas Yau

**Affiliations:** 1Department of Medicine, Queen Mary Hospital, The University of Hong Kong, Hong Kong, China; yawen.dong@gesundheitsverbund.at (Y.D.); wsl714@ha.org.hk (J.S.L.W.); bryanli@hku.hk (B.L.); kgw951@ha.org.hk (G.G.W.K.); lcy035@ha.org.hk (R.L.); jwychiu@hku.hk (J.W.Y.C.); 2Department of Surgery, Klinik Favoriten, Wiener Gesundheitsverbund, 1100 Vienna, Austria; 3School of Biomedical Science, The University of Hong Kong, Hong Kong, China; rios@hku.hk; 4Department of Clinical Oncology, Queen Mary Hospital, The University of Hong Kong, Hong Kong, China; lamkaon@hku.hk; 5Department of Surgery, Queen Mary Hospital, The University of Hong Kong, Hong Kong, China; cheung68@hku.hk

**Keywords:** hepatocellular carcinoma, biomarker, checkpoint inhibitors, combination therapy, targeted therapy, precision medicine, ICI response, transcriptomics

## Abstract

**Simple Summary:**

Hepatocellular carcinoma is an aggressive cancer with high mortality. The introduction of immune-checkpoint inhibitors (ICIs) targeting the programmed-cell death 1 pathway (PD-1/L1) has led to a paradigmatic change in the systemic treatment of HCC. However, the important challenges of how to predict who would respond to ICIs and how to further optimize treatment remain. Strategies are being explored using anti-PD-1/L1 as a staple in combination with other agents to improve outcomes. This includes, but is not limited to, targeting other immune-checkpoints and alternative pathways involved in HCC development. Furthermore, multiple biomarkers are under investigation to predict ICI responders. We discuss available studies and future prospects of ICI use in HCCs in these two directions.

**Abstract:**

Advanced, unresectable hepatocellular carcinoma has a dismal outcome. Multiple immune checkpoint inhibitors (ICIs) targeting the programmed-cell death 1 pathway (PD-1/L1) have been approved for the treatment of advanced HCC. However, outcomes remain undesirable and unpredictable on a patient-to-patient basis. The combination of anti-PD-1/L1 with alternative agents, chiefly cytotoxic T-lymphocyte antigen-4 (CTLA-4) ICIs or agents targeting other oncogenic pathways such as the vascular endothelial growth factor (VEGF) pathway and the c-MET pathway, has, in addition to the benefit of directly targeting alterative oncogenic pathways, in vitro evidence of synergism through altering the genomic and function signatures of T cells and expression of immune checkpoints. Several trials have been completed or are underway evaluating such combinations. Finally, studies utilizing transcriptomics and organoids are underway to establish biomarkers to predict ICI response. This review aims to discuss the biological rationale and clinical advances in ICI-based combinations in HCCs, as well as the progress and prospects of the search for the aforementioned biomarkers in ICI treatment of HCC.

## 1. Introduction

Accounting for up to 90% of all primary liver malignancy, hepatocellular carcinoma (HCC) is a highly aggressive cancer type and the fourth most common cause of cancer-related deaths worldwide [1]. HCC develops within an immune-tolerant niche, as the liver has multiple distinct mechanisms to suppress unwanted immune activation from bypassing antigen and bacteria channeled through the portal vein system, including the upregulation of immune checkpoints such as the programmed death-1 (PD-1) pathway [2,3]. In addition, HCC is nearly always accompanied by chronic hepatic inflammation and cirrhosis. Consequently, HCCs have been theorized, and indeed proven, to be responsive to immune-checkpoint inhibitors (ICIs). The first single agent ICI to be approved in HCC was the anti-PD-1 nivolumab, which demonstrated an objective response rate (ORR) of 15% in the phase I/II CheckMate-040 trial [4]. In addition, pembrolizumab, another anti-PD-1, was approved as a single agent with the results of the phase I/II Keynote-224 trial, which demonstrated an ORR of 17% [5]. However, in spite of the successful phase I/II trials of nivolumab and pembrolizumab, both agents did not reach their primary endpoints in their phase III trials (Checkmate-459 and Keynote-240), demonstrating a median OS of 16.4 months and 13.9 months, respectively [6,7].

The definite but limited clinical benefit of single agent anti-PD-1 gave fresh impetus to the search for strategies to optimize ICI treatment. These include attempting to enhance response by combining ICIs with other ICIs, tyrosine kinase inhibitors (TKIs), anti-vascular endothelial growth factor antibodies (anti-VEGFs), and other agents. In addition, biomarkers are currently being searched for to predict responders. In this review, the biological basis, clinical data, and further development of ICI combinations in HCC, as well as the existing literature and future prospects of biomarkers to predict ICI responders will be discussed.

## 2. Immune Checkpoints and the Programmed Death-1 Pathway

In an adaptive immune response, two signals are required for T cell activation. Through MHC peptides, antigen-presenting cells present unique antigens and provide specificity to T cell response. A co-regulatory signal then either stimulates or inhibits T cell activation. Immune checkpoints are intrinsic to the immune system to prevent autoreactivity. Amongst the most studied immune checkpoints are the programmed death-1 pathway (PD-1) and the cytotoxic T-lymphocyte associated protein-4 (CTLA-4) pathway. Knocking out PD-1 in mice leads to the development of diverse autoreactivity phenomena including type 1 diabetes mellitus and autoimmune encephalomyelitis [8,9]. Meanwhile, knockout of CTLA-4 in mice leads to massive and fatal lymphoproliferation [10]. However, these pathways are often upregulated in the tumor microenvironment (TME), leading to T cell suppression and tumor immune escape. Last but not least, a variety of other immune co-regulatory signals have been studied in the context of tumor immune response (Figure 1).

The programmed death-1 pathway consists of the programmed cell death protein 1 (PD-1) and programmed death ligand 1 (PD-L1). PD-1 is an immune co-inhibitory receptor expressed during inflammation or infection on CD8+ T cells and other immune cells [11]. By contrast, PD-L1, the primary ligand of PD-1, is constitutively expressed on the surface of antigen-presenting cells (APCs) as well as other tissues, including on tumor cells [12]. Upon binding of PD-L1 to its receptor PD-1, the PD-1/PD-L1 signaling pathway is activated, resulting in a suppression of antigen-specific T cell activation through a variety of mechanisms [13]. Several pathways have been implicated in the upregulation of PD-L1 expression in cancer. An upregulation of PD-L1 messenger RNA transcription has been attributed to a loss-of-function mutation of the phosphatase and tensin homolog (PTEN) [14]. The activation of the transcription factor STAT3, as well as the NF-kB pathway can also contribute to an enhanced PD-L1 mRNA transcription [15,16]. In addition, activated T cells facilitate PD-L1 expression by releasing distinct pro-inflammatory proteins during an immune response, which have a positive impact on the transcription of PD-L1 mRNA [17]. The most prominent pro-inflammatory molecule that has been ascribed to this process is interferon-gamma (IFN-γ), which is secreted by tumor-associated antigen specific T cells in the TME. Upon interaction of IFN-γ with its receptor (IFNGR) on tumor cells, the downstream Jak/Stat signaling cascade is triggered, resulting in an upregulation of PD-L1 transcription (Figure 2). Anti-PD-1/L1 antibodies block either PD-1 or PD-L1 to increase CD8+ T cell proliferation and partially reverse T cell anergy, resulting in increased anti-tumor immune response [13]. 

## 3. Immune Checkpoint Inhibitor Combinations

### 3.1. Anti-PD-1/L1 and Anti-CTLA-4

Apart from the PD-1 pathway, CTLA-4 is another immune checkpoint that has been successfully targeted. Unlike PD-1 which suppresses T cells at a later stage in the immune response and in peripheral tissues, CTLA-4 mediates T cell proliferation early in an immune response and within lymph nodes [12]. It is a co-inhibitory molecule which competes with the co-stimulatory CD28 for binding to CD80/86. By having a higher affinity for CD80/86 than CD28, CTLA-4 effectively impairs the activation of effector T cells. Furthermore, CTLA-4 is expressed constitutively on regulatory T cells (Tregs) and thus plays an essential role in the immunosuppressive activity of Tregs in the TME. Consequently, ICIs targeting the CTLA-4 pathway aims to counter-act the ability of tumor cells to escape the host immune surveillance by promoting and reactivating tumor-specific T cells (Figure 3). Combined blockade of PD-1/L1 and CTLA-4 offers several distinct advantages over single PD-1 pathway blockade. Firstly, blockade of PD-1/L1 is associated with upregulation of CTLA-4, and thus a direct additional blockade may prevent further immune escape [18]. Secondly, myeloid-derived suppressor cells (MDSCs) can severely restrict T cell activity within the TME, but combined blockade can synergistically increase the proportion of CD8+ effector T cells relative to MDSCs [19]. Thirdly, combined blockade increases production of inflammatory cytokines such as TNF-alpha and IFN-gamma and decreases T cell anergy [19]. Finally, combined blockade can lead to the expansion of memory T cells (Tmem), which can confer longer-term anti-tumor immunity compared to effector T-cells [20,21]. In fact, in melanoma patients with exceptional responses to ICIs, Tmem have been known to persist both in skin and blood for up to a remarkable 9 years. The persistent Tmem also preferentially expressed a high IFN-gamma/TNF signature, which strongly prognosticates melanoma patients [22]. These findings suggest that Tmem are critical in long-term anti-tumor immunity, which anti-CTLA-4 is uniquely positioned to enhance. Consequently, anti CTLA-4 and anti-PD-1/L1 have been established in multiple tumor types such as melanoma, lung cancer, urothelial carcinoma, renal cell carcinoma, esophagogastric cancer, and MSI-high colorectal cancer [23,24,25,26,27,28,29].

In HCC, two CTLA-4 ICIs, ipilimumab and tremelimumab, have been evaluated in combination with anti-PD-1/L1 agents. Ipilimumab–nivolumab is the only combination-ICI regime to be approved by the US FDA for use in HCC so far, having been approved in sorafenib-treated advanced HCC patients based on the results from cohort 4 of the phase I/II CheckMate-040 study [30]. In this study, ipilimumab–nivolumab was given to a total of 148 patients in one of three treatment arms: arm A with nivolumab 1 mg/kg + ipilimumab 3 mg/kg every 3 weeks for 4 doses; arm B with nivolumab 3 mg/kg + ipilimumab 1 mg/kg every 3 weeks for 4 doses, with each arm being followed by single agent nivolumab 240 mg every 2 weeks; or arm C with nivolumab 3 mg/kg Q2W + ipilimumab 1 mg/kg every 6 weeks continuously. In terms of clinical benefit, the ORRs were 32% in arm A, 31% in arm B, and 31% in arm C, the CR rates were 8%, 6%, and 0%, and the median OS was 22.8, 12.5, and 12.7 months, respectively. Recently published long term results showed a similar median OS, with the 3-year survival rates of arms A, B, and C being 42, 26, and 30 months, respectively [31]. In general, ipilimumab–nivolumab was well tolerated despite having higher rates of adverse events (AEs) compared to nivolumab monotherapy. All grade and grade ≥3 AEs were 94% and 55% in arm A, 71% and 29% in arm B, and 79% and 31% in arm C. Of note, although the study was not powered to compare outcomes across different arms, patients in arm A, who had the highest ipilimumab dose, had the longest median overall survival and highest long term survival rates, but also the highest incidence of AEs. This interesting phenomenon has been further explored in a dose exposure–response analysis based on the trial population. Using population pharmacokinetics, ipilimumab and nivolumab exposure were generated and outcomes analyzed [32]. High ipilimumab exposure, especially for patients in arm A, but not nivolumab exposure, was associated with better overall survival. Overall, the results of the CheckMate-040 trial, especially for arm A, compares favorable against other treatment options in the post-sorafenib setting. Of particular note is that although cross-trial comparisons are difficult, the median OS of arm A represents the longest amongst all regimes approved in this setting. Although the overall toxicity was higher, the overall profile of AEs was similar to other ICIs. About 50% of the patients in arm A needed systemic steroids treatment. However, immune-mediated AEs (IRAE) mostly resolved across all treatment arms, and intriguingly, for patients who were rechallenged after experiencing an IRAE, few experienced recurrence of the event. Appendix A describes the exclusion criteria of ICI combination regimens.

Aside from ipilimumab, tremelimumab has also been evaluated with durvalumab (an anti-PD-L1) in advanced HCC. In the phase I/II Study 22, tremelimumab was investigated either in direct combination with durvalumab, or as a one-off priming therapy to enhance the anti-tumor activity of durvalumab without significant added toxicity [33]. In this study, patients who were previously treated with sorafenib were randomized to one of the following arms: arm A = tremelimumab 300 mg (1 dose) plus durvalumab 1500 mg every 4 weeks; arm B = tremelimumab 75 mg (4 doses) plus durvalumab 1500 mg every 4 weeks; arm C = durvalumab monotherapy 1500 mg every 4 weeks; arm D = tremelimumab monotherapy 750 mg every 4 weeks (7 doses, then every 12 weeks thereafter). In general, patients in arm A had better outcomes, such as a higher ORR (arm A, 24%; arm B, 9.5%; arm C, 10.6%; arm D, 7.2%) and a longer median OS (arm A, 18.7 months; arm B, 11.3 months; arm C, 13.6 months; arm D, 15.1 months). In terms of safety profile, all grade treatment-related AEs (TRAEs) and grade ≥3 TRAEs occurred in 82.4% and 35.1% of patients in arm A, 69.5% and 23.2% in arm B, 60.4% and 17.8% in arm C, and 84.1% and 43.5% in arm D. Specifically, the incidence of TRAEs requiring systemic steroids for arm A was 24.3%, and the safety profile appeared favorable. Interestingly, the proliferative CD8+Ki67+ T-cell count was associated with radiological response regardless of treatment received but was highest for arm A, supporting the proposed mechanism of enhanced immune activation with higher dose tremelimumab.

Based on the encouraging results of the CheckMate-040 cohort 4 and Study 22, ipilimumab–nivolumab and tremelimumab–durvalumab are both currently being investigated in phase III trials as first-line treatment in advanced HCC. The CheckMate-9DW study aims to compare ipilimumab–nivolumab to lenvatinib or sorafenib, while the HIMALAYA study aims to compare tremelimumab–durvalumab either as combination or with tremelimumab as a priming agent (as per Study 22) with durvalumab monotherapy or sorafenib [34,35]. The results of these trials would further define the role of anti-CTLA-4 and anti-PD-1/L1 combinations in different settings in advanced HCC.

Apart from being used as combination or priming therapy, anti-CTLA-4 has also been studied as salvage therapy for patients who are refractory to anti-PD-1/L1. We recently published the first study on the use of ipilimumab–nivolumab/pembrolizumab in patients with PD-1 pathway ICI refractory HCC [36]. All 25 patients received ipilimumab 1 mg/kg with nivolumab 3 mg/kg or pembrolizumab 2 mg/kg every 3 weeks. Importantly, the ORR was 16%, CR rate was 12%, and median OS was 10.9 months. There was no difference in ORR between those with prior response to ICIs compared to those without. Prospective trials will be conducted to validate the addition of anti-CTLA-4 for those with HCCs refractory to PD-1 pathway ICI, which is an expanding and ever more important patient population.

### 3.2. ICIs and TKIs/Anti-VEGFs

Aside from ICIs, the other major group of systemic HCC therapeutics are tyrosine kinase inhibitors (TKI)/anti-VEGF monoclonal antibodies. Approved agents include the TKIs sorafenib and lenvatinib as first line treatment, as well as regorafenib and cabozantinib as post-sorafenib second-line treatment. In addition, the anti-VEGF-2 monoclonal antibody ramucirumab is approved for use in post sorafenib patients with alpha-fetoprotein ≥400 ng/mL (Figure 4). All such agents target the VEGF pathway, long implicated in HCC pathogenesis and progression [37]. In addition, individual TKIs target other pathways such as FGF receptors (1–4), KIT, platelet-derived growth factor receptor alpha, c-MET, RET, and B-RAF pathways. Although all these agents have been licensed as monotherapy, there is ample in vitro evidence to support their use with ICIs for synergistic effect [38]. VEGF may downregulate adhesion molecules needed for T cell trafficking such as ICAM-1 in the TME. An increased VEGF level is also associated with increased tumor vessel formation, which is paradoxically associated with increased hypoxia and acidosis. This hypoxic TME leads to recruitment of immunosuppressive cells such as Tregs and MDSCs. Blockade of such pathways thus improves T cell infiltration and activity in the TME. Individual TKIs may additionally have specific immunomodulatory effects. Sorafenib and lenvatinib have been shown to be able to either deplete or alter the phenotype of tumor-associated macrophages (TAMs) from a M2 polarized immunosuppressive state to a M1, pro-anti-tumor immunity state [38,39,40]. Regorafenib has been shown to increase T cell cytokine production, reduce co-inhibitory receptor expression, and increase the expression of CD25 and CD28 on T cells [41]. Finally, cabozantinib specifically blocks the c-MET pathway, which, in addition to being a direct pathogenesis pathway for HCC, mediates direct phosphorylation and activation of GSK3B, causing decreased expression of PDL1 [42,43]. Based on these results, multiple ICI and TKI/anti-VEGF combinations have been considered in the treatment of HCCs (Figure 5 and Figure 6).

In the landmark IMBrave150 trial (the first ever successful phase III trial of ICIs in advanced HCC), atezolizumab (an anti-PD-L1) combined with bevacizumab (an anti-VEGF monoclonal antibody) was compared against sorafenib in the first-line setting. A total of 501 patients were randomized in a 2:1 ratio to atezolizumab–bevacizumab or sorafenib. In the primary analysis, the atezolizumab–bevacizumab arm had superior overall survival (hazard ratio for death 0.58, 95% CI 0.42–0.79, *p* < 0.001), progression free survival (hazard ratio for disease progression or death 0.59, 95% CI 0.47–0.76, *p* < 0.001), and ORR by RECIST 1.1 (27.3% vs. 11.9%, *p* < 0.001) compared to the sorafenib arm. AEs of any grade, grade 3–4, and grade 5 were reported in 98.2%, 56.5%, and 4.6% in the atezolizumab–bevacizumab arm, and 98.7%, 55.1%, and 5.8% in the sorafenib arm. Most frequent AEs were those related to anti-VEGF such as hypertension [45]. A follow-up long-term analysis showed a median OS 19.2 months of the atezolizumab–bevacizumab arm vs. 13.4 months of the sorafenib (*p* = 0.0009). Survival at 18 months was 52% with atezolizumab–bevacizumab and 40% with sorafenib. No new safety signals were identified [46]. The success of IMbrave150 has several significant impacts on the treatment landscape of HCC: firstly, atezolizumab–bevacizumab has now been established as arguable the first-line treatment of choice; secondly, ICI and VEGF targeting agent combinations have been proven to be safe and efficacious; and finally, future clinical research of second-line treatment should be focused on anti-PD-1/L1 resistant patients.

Besides atezolizumab–bevacizumab, ICI has also been evaluated with TKIs in advanced HCC. In the phase I/II CheckMate-040 cohort 6 study, the efficacy and safety of nivolumab–cabozantinib with or without ipilimumab (doublet and triplet) in sorafenib-naïve or experienced patients were evaluated [47]. The ORR was 17% and 26%, and disease control rate (DCR) was 81% and 83% for the doublet and triplet arms, respectively. Median PFS was 5.5 mo for the doublet arm and 6.8 mo for the triplet arm, whereas median OS was not reached in either treatment group. Nevertheless, more frequent Grade 3–4 TRAEs were reported, with the incidence reaching 42% and 71% of patients in the doublet and triplet arms, respectively. In addition, the combination of lenvatinib and pembrolizumab was evaluated in the phase Ib KEYNOTE-524 trial [48]. Most patients had no prior systemic therapy. By RECIST 1.1, an ORR of 36% and median PFS of 8.6 months was achieved. The median OS was impressive (22.0 months). In total, 99% and 67% of patients experienced any and grade ≥3 AEs, respectively. Camrelizumab, another anti-PD-1, was evaluated with the anti-VEGF-2 TKI apatinib in the RESCUE trial in both first and second line (to TKI) settings. In total, 75 first line and 120 s line patients were enrolled. The ORR was 34.3% in the first line and 22.5% in the second line. Median PFS was 5.7 and 5.5 months, and 12-month survival rates were 74.7% and 68.2%, respectively. Grade ≥3 TRAEs occurred in 77.4% of patients [49]. Finally, the VEGF Liver 100 trial evaluated the anti-PD-L1 avelumab in combination with axitinib (VEGF receptor 1/2/3 inhibitor) in treatment-naïve advanced HCC patients [50]. Out of 22 patients, the ORR was 13.6% by RECIST 1.1 (31.8% by mRECIST) and median PFS was 5.5 months by RECIST 1.1. The OS data were yet to be mature.

In the setting of advanced HCC, three ongoing phase III trials evaluating ICI-TKI combinations in the first line are underway. The COSMIC-312 trial aims to evaluate the atezolizumab–cabozantinib combination compared to sorafenib and cabozantinib monotherapies [51]. Additionally, pembrolizumab–lenvatinib is being evaluated in the LEAP-002 study compared against lenvatinib monotherapy [52]. Finally, camrelizumab is being evaluated with apatinib compared to sorafenib [53]. The results of these trials would enlighten the HCC community as to the synergism between TKIs targeting multiple pathways and ICIs.

In general, only limited real-world data on the use of ICI-TKI combinations exist. We recently reported the use of cabozantinib with ICIs in 11 patients, with 45.5% having stable disease and a median OS of 12.3 months [54]. ICI-TKIs have also been investigated as a potential treatment strategy to downstage advanced HCC to resectable HCC. In a recently published real-world study, 63 patients with advanced HCC received either lenvatinib– pembrolizumab or camrelizumab–apatinib. In total, 15.9% of patients were able to receive curative resection following systemic treatment [55]. 

### 3.3. Anti-PD-1 and Anti-LAG3

Lymphocyte activation gene-3 (LAG3) is the third and latest immune checkpoint which has garnered sufficient interest to be exploited therapeutically. LAG3 has diverse but incompletely understood functions both intracellularly and extrinsically [56,57]. It is expressed on CD4+ and CD8+ T cells as well as NK cells during activation phase [58]. Overall, LAG3 is a negative regulator of T cell activation and function, as well as a key contributor to T cell exhaustion. Knockout of LAG3 in mouse models leads to deficiencies in cell cycle arrest/death of T cells, splenomegaly, and autoreactive phenomena such as accelerated diabetes [59,60]. In vitro studies show that blockade of LAG3 leads to enhanced proliferation of CD4+ and CD8+ T cells and increase inflammatory cytokine release such as IFNγ [61,62,63,64]. Converse to its expression on effector T cells and NK cells, LAG3 is constitutively expressed on thymic-derived Tregs [65]. It has been shown to promotes Treg differentiation and suppressive activity, and LAG3 blockade both inhibits Treg induction and proliferation [65,66]. Finally, LAG3 has a number of inhibitory functions on dendritic cells and NK cells, although further studies are needed to map out such pathways [56].

LAG3 is commonly co-expressed with other immune checkpoints such as PD-1, resulting in an anergic, dysfunctional T cell state [56]. Interestingly, single target blockade of LAG3 had a limited effect on both resolution of chronic infection and tumor clearance in mouse models with exhausted T cells. However, combination blockade with anti-LAG3 and anti-PD1 dramatically increased viral and tumor clearance [67,68]. Based on these results, the first-in-class anti-LAG3 relatlimab is currently being evaluated in a number of solid-organ tumors in various trials including in HCC [69,70]. The first results, from the RELATIVITY-047 trial of nivolumab–relatlimab in treatment naïve melanoma, met its primary endpoint of progression-free survival [71]. The details of the RELATIVITY-047 trial and the results of other ongoing trials will further clarify the role of anti-LAG3 in HCC and other solid organ tumors.

### 3.4. ICI and Chemotherapy

Chemotherapy is well known to have immunomodulatory effect on cancers. This is achieved through multiple mechanisms: firstly, certain chemotherapy agents such as oxaliplatin mediate the release of tumor antigens and enhance signaling for phagocytosis such as CRT and HMGB1. This activates the NRLP3 inflammasome, leading to the production of IL1 beta, dendritic cell maturation, and eventually, CD8+ T cell recruitment and activation [72,73]. Furthermore, chemotherapy can upregulate tumor antigens directly, as well as upregulate co-stimulatory molecules and downregulate immune checkpoints [72,74]. Finally, chemotherapy has been known to restrict Treg and MDSC activities, thus reducing immunosuppression and enhancing anti-tumor immunity [75,76]. Therefore, chemotherapy and ICIs represent another promising option to achieve synergistic effect. To date, one trial exists investigating ICI–chemotherapy combinations, which is a phase II study containing 34 HCC patients who received camrelizumab with FOLFOX (5-Fluorouracil, leucovorin, and oxaliplatin) or GEMOX (gemcitabine–oxaliplatin) as first line therapy [77]. The ORR was 26.5% and DCR was 79.4%. Median PFS was 5.5 months with median OS not reached. Grade ≥3 TRAEs occurred in 85.3% of patients. A phase III study is underway to investigate camrelizumab with FOLFOX in treatment for naïve advanced HCC patients [78].

### 3.5. ICI and Locoregional Therapies or Radiotherapy

Locoregional therapies such as transarterial chemoembolization (TACE) and ablation have been shown to stimulate tumor antigen specific CD8+ T cells, Tmem, and NK cell responses [79,80,81]. Based on these findings, a number of locoregional therapies have been evaluated in combination with ICIs. In a phase I/II single arm study, tremelimumab was evaluated in combination with TACE or RFA for treatment of 32 patients with advanced HCC [82]. In total, 19 patients had evaluable response with 5 (26.3%) having partial response. Interestingly, a significant increase in CD3+, CD4+, and CD8+ T cells was observed in peripheral blood and intratumorally after treatment, with responders having a higher number of CD3+/CD8+. Furthermore, in the published results of the PETAL study, 6 patients were given pembrolizumab following TACE. Of 4 patients with evaluable responses, 3 had stable disease. No synergistic toxicity was reported [83]. Several ongoing phase III trials are underway evaluating ICIs with TACE, including CheckMate-74W, EMERALD-1, and LEAP-012 trials. In these 3 studies, ipilimumab–nivolumab or nivolumab, durvalumab–bevacizumab or durvalumab, and pembrolizumab–lenvatinib are being evaluated against TACE, respectively. The results of these trials will further shine light on whether ICIs can be used effectively beyond the advanced HCC setting [84,85,86]. 

Finally, radiotherapy (RT) may play a role in overcoming ICI resistance due to augmented tumor antigen release, T cell activation/ infiltration, and MHC class I expression [87]. Furthermore, ICIs may enhance the well-known abscopal effect of RT: the immune-mediated phenomenon of tumor regression outside of the irradiated field. To date, two published trials combining ICIs with RT exist. In a phase II single center study, Y-90 radioembolization was combined with nivolumab for treatment of 40 patients. Of 36 evaluable patients, the ORR was 31% and DCR was 58.3% The median PFS and OS were 4.6 months and 15.1 months, respectively. In total, 11% of patients had ≥grade 3 treatment-related AEs (TRAEs) [88]. In another phase II trial including 42 patients, who were treated with selective internal radiation therapy (SIRT) with subsequent nivolumab, the ORR was 38%, DCR was 81%, time-to-progression (TTP) was 9.3 months, and median OS was 20.6 months. Grade ≥3 TRAEs were found in 48% of patients [89]. Several other phase II clinical trials investigating the combined approach of ICIs and radiotherapy are underway, including the combined use of durvalumab–tremelimumab with RT and stereotactic body RT (SBRT) with ipilimumab [90,91]. 

## 4. Precision Medicine of ICI

The fundamental question in ICI lies in how to predict response in each patient. The precision medicine of ICI depends on the identification of biomarkers that predict its response. The thorough transcriptomic analysis of the tumor immune microenvironment has identified several key inflammatory features in other cancers. Further investigation of these features in HCC will lead to the useful biomarkers of ICI response. 

### 4.1. Potential Biomarkers of Clinical Response in ICI

The CheckMate-040 trial defined inflammatory signatures as potential biomarkers of ICI responses [92]. From biopsy of tumors and blood drawn from the patients with HCC, ICI responders upregulated the expression of 4 inflammatory genes (PD-L1, CD8A, LAG3, STAT1). On the contrary, the expression of markers of both T-cells (CD4 and CD8) and macrophages did not correlate with overall survival. This indicates the need for higher resolution profiling of immune cell types for the identification of useful biomarkers.

The identification of ICI responder-unique immune cells types will boost the discovery of biomarkers. The single-cell resolution analysis provided a clear insight on the tumor microenvironment in response to ICI. A new study described the response to ICI in renal cell carcinoma (RCC) with single-cell RNA-seq [93]. Eight patients with advanced RCC were recruited in the study. Five patients underwent ICI, and three patients did not. Biopsy samples of all patients underwent single-cell RNA-seq analysis and were categorized as ICI-naïve, responder, and non-responder according to clinical response. ICI changed a subset of CD8+ T-cells defined as ‘4-1BB lo’ to express a high level of both inflammatory effector molecules (IFNG, PRF1) and co-inhibitory molecules (PD-1, LAG3). Tumor-associated macrophages shifted to an inflammatory M1-like phenotype with high level expression of co-inhibitory molecules (VSIR, VSIG4, PD-L2, SIGLEC-10). These observations suggest that the tumor microenvironment may adapt to ICI treatment by increasing immunosuppressive gene expression.

Another study in melanoma patients revealed a specific subset of T-cells correlated with ICI responders [94]. Twenty-eight patients with melanoma received anti-PD-1 ICI. Single-cell RNA-seq of drawn blood identified a high proportion of activated effector memory CD8+T-cells in responders. Further computational analysis of these T-cell subsets defined mucosal-associated invariant T (MAIT) cells with homing properties and expressed a high level of KLRB1, SLC4A10, MAF, and CXCR4. Intriguingly, the level of MAIT cells before therapy can predict the initial response to ICI. However, another study suggested that MAIT promoted tumor initiation in the melanoma xenograft murine model [95]. Therefore, the presence and the role of MAIT need thorough analyses in HCC.

Besides transcriptional characterization, cellular metabolomics at the single-cell level defined the rare early activated CD8+T-cells in a primary immune response [96]. A mass cytometry approach revealed that early activated T-cells exhibited simultaneous peaks in both glycolysis and OXPHOS, which are distinct from naive and committed T-cells. Intriguingly, a similar early activated population was identified in CAR-T cells upon infusion in lymphoma patients, indicating the dual metabolic peaks as potential biomarkers of responding T-cells in ICI.

### 4.2. The Future Platform of Translational Research

The further investigation of immune-cancer crosstalk will lead to new biomarkers of ICI. The progress of single-cell transcriptomics will advance the discovery of biomarkers. Though conventional single-cell RNA-seq lack precise information of immune-cancer crosstalk, the recent inventions of spatial transcriptomics would push the limit of this technology. Slide-seq is a method for transferring RNA from tissue sections onto a surface covered in DNA-barcoded beads with known positions, which allows the detection of RNAs regionally. Spatially resolved gene expression data at resolutions comparable to the single-cell level were obtained in both cerebellum and hippocampus [97]. Stereo-seq combined DNA nanoball patterned array chips and in situ RNA capture. This technology allowed high sample throughput transcriptomic profiling of histological sections at nanoscale resolution with areas up to a centimeter scale. The almost single-cell resolution of spatial transcriptomics was applied to an adult mouse brain and sagittal sections of E11.5 and E16.5 mouse embryos [98]. 

The lack of a proper experimental model of immune-cancer crosstalk in HCC has hampered our ability to discover novel biomarkers of ICI. For example, CRISPR-based screening and lineage tracing had identified key factors to determine chemotherapy resistance and metastasis of cancer, while missing in the context of ICI [99,100]. The use of 2D cell lines and 3D microfluidic models of HCC cannot fully recapitulate the tumor microenvironment. Patient-derived xenografts are missing human immune cells that play key roles in ICI [101]. Recent progress in HCC organoids has advanced our understanding of drug screening on the tumor [102]. A study testing drug screen compounds identified an ERK inhibitor, SCH772984, as a potential therapeutic agent for patients with primary HCC [103]. Although HCC organoids can be established using a biopsy sample, immune cells are missing in these HCC organoids, presenting challenges for modeling ICI response [104].

The assembly of HCC organoids with complement immune cells will advance our understanding of the biomarkers of ICI response. Assembloids of HCC organoids combined with immune cells allow immune-cancer crosstalk in response to ICI. The lack of immune cells in HCC organoids will be overcome by the addition of immune cells afterward of the organoid formation. Previously, two or more organoids representative of different brain region identities were fused to form assembloids to model interactions between different brain regions [105]. This technology was adopted in cancer research by co-culturing peripheral blood-derived immune cells (PBMCs), which enacted screening of drugs that target melanoma and immune cell crosstalk [106]. 

The precise experimental platforms that capture immune-cancer crosstalk will advance the identification of new biomarkers to detect ICI response. The combination of spatial transcriptomics and assembloid technology would lead to the discovery of biomarkers. Furthermore, CRISPR-based genetic screening would define novel biomarkers that determine the prognosis of ICI (Figure 7). 

### 4.3. The Future Development of ICI Combinations

The future development of ICI combination is likely to proceed in the following directions: new combination agents and diversified settings. 

Firstly, the mechanisms of different combination regimes should be illustrated, and new agents evaluated. With the exception of sorafenib, the immunomodulatory effects of individual TKIs have arguably been at most incompletely understood [38]. Better understanding may pave the way to more optimized combination with different ICIs (e.g., certain TKIs may benefit from the addition of CTLA-4 whilst others may only benefit from anti-PD-1/L1 combinations). Special focus should be placed on the triple combination of anti-PD-1/L1, anti-CTLA-4, and anti-VEGF, as this combination has the broadest possible blockade out of all currently established HCC treatment options and thus potentially strongest synergism and anti-tumor effect. 

The variation in ICIs is potentially no less important than that of the combination partner. There appears to be a difference in outcomes for various PD-1 pathway ICIs despite being trialed in the same settings [107]. One possible explanation is that different ICIs may stimulate the production of antidrug antibodies (ADA) by the humoral immune system to a different degree. To date, the ICI with the highest incidence of, and most well-studied ADA phenomena is atezolizumab, which has been quoted to have up to 54.1% of patients developing ADAs in selected analysis [107]. The incidence of, and indeed implications of ADA, is not conclusive—some studies suggest that ADAs are neutralizing and thus reduce efficacies of ICIs, others suggest that they either have no impact or even positive relationships with survival outcomes [107,108,109]. Future studies may be able to define the role of ADAs and relative efficacies of different ICIs in HCC.

In addition to the aforementioned LAG3, newer treatment targets and modalities should be evaluated in combination with existing ICI/TKIs. Numerous exciting possibilities exist. Blockade of T-cell immunoglobulin mucin-3 (TIM-3), an alternative immune checkpoint which is upregulated in PD-1 blockaded tumors and also in tumor-infiltrating lymphocytes, has been shown to overcome resistance to PD-1 blockade [18]. Furthermore, high-affinity NK cells (haNK), which can be engineered to target PD-L1, have been shown to direct anti-tumor effects and suppressive MDSCs [110]. Indeed, a trial of t-haNK in pancreatic cancer is underway after early phase studies showed haNK to have an encouraging tumor response in combination with avelumab [111,112,113]. Chimeric antigen receptor T-cell therapy (CAR-T) has also shown promise in HCC, with several phase I studies reporting promising anti-tumor activities recently [114]. 

Secondly, ICI combinations should be evaluated in different settings. The breakneck speed of ICI development has presented researchers across different tumor types with the common problem of often having designed studies with outdated comparison arms or earlier-line treatments. For example, the IMbrave150 study has established atezolizumab–bevacizumab as arguably the new first-line treatment of choice, but newer and ongoing trials are still using sorafenib as the comparison arm or as first-line. For this reason, ICI combinations need to be evaluated in a broader range of settings, with perhaps none more important than in post-ICI patients. As mentioned above, we evaluated the addition of ipilimumab to anti-PD-1 in PD-1 pathway ICI refractory patients. Further possibilities include the use of add-on TKIs/VEGFs to ICIs after ICI progression and locoregional therapies/RT to enhance ICI response after ICI exposure. Different resistance patterns and mechanisms should also be mapped to design therapies best suited to restore response. In addition, the use of ICI combinations in patients with advanced cirrhosis should also be evaluated. The majority of HCC patients have concomitant cirrhosis and thus a large portion of real-world patients would not be eligible to receive ICIs based on clinical trial inclusion criteria. Therefore, data on the use of ICI and ICI combinations in such patients is deficient. Large and unmet medical needs in these areas exist and need further study. Last but not least, the efficacies of ICIs in HCCs of different etiologies should be clarified, in particular in non-alcoholic steatohepatitis (NASH) [115]. In a recent meta-analysis of the results of the CheckMate-459, IMbrave150, and Keynote-240 trials, non-viral (alcoholic or NASH) patients had no superior survival compared to the control arms, unlike HBV/HCV-related HCC patients [115]. Furthermore, non-alcoholic fatty liver disease (NAFLD) was independently associated with shortened median OS after immunotherapy. In preclinical models, NASH-affected livers had exhausted, activated CD+PD1+ T cells accumulated, which expanded after PD-1 ICI but did not lead to tumor regression. When given prophylactically, anti-PD1 actually increased the incidence and extent of NASH-HCC. These results are especially intriguing as NAFLD is both highly prevalent and an increasingly common etiology for HCC [116]. In addition, NAFLD is commonly coexistent with chronic hepatitis B, and has diverse effects on fibrosis progression in chronic hepatitis B [117]. Thus, the role of ICI and ICI combinations in HCCs arising from the background of NASH-predominant or NAFLD-affected livers require further prospective study.

## 5. Conclusions

The advent of ICIs has led to tremendous progress in the treatment of HCC. Combination treatment with other ICIs, TKIs/anti-VEGFs, chemotherapy, locoregional therapies, and RT are promising approaches to enhance anti-tumor efficacy. Additionally, biomarkers predicting responders are being searched for to optimize treatment. Further studies are needed to advance the use of ICI combinations in HCC along these directions.

## Figures and Tables

**Figure 1 cancers-13-01949-f001:**
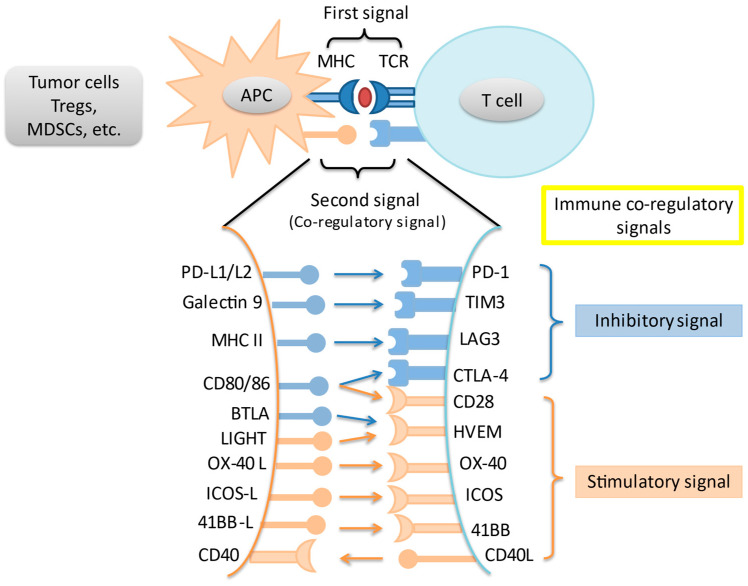
Interaction of main immune co-stimulatory/co-inhibitory molecules and their corresponding receptors. Cancer cells and other cells in the tumor microenvironment express a variety of inhibitory and stimulatory ligands that bind to their cognate receptors on immune cells, thereby leading to immune-modulation. These ligand-receptor pairs are known as immune checkpoints. Tregs = regulatory T cells; MDSC = myeloid-derived suppressor cell; APC = antigen presenting cell; TCR = T-cell receptor.

**Figure 2 cancers-13-01949-f002:**
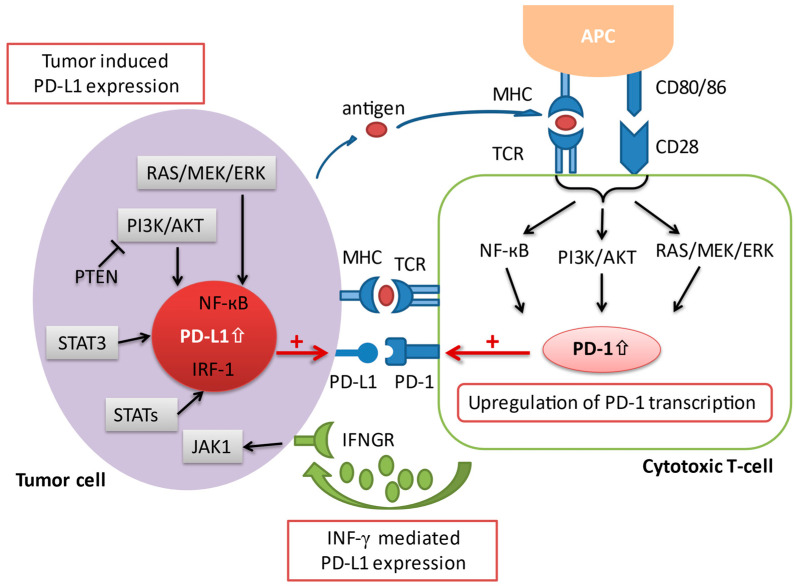
A schematic overview illustrating the various aspects involved in the regulation of PD-1/PD-L1 transcription. Co-stimulation via CD28 in conjunction with the TCR initiates the activation of several signaling pathways, including the Ras/MEK/ERK, NF-κB, and PI3K/AKT cascade, which in turn leads to an upregulation of PD-1 transcription. On the other hand, the expression of PD-L1 on tumor cells is regulated via 1) oncogenic activation of different signaling pathways and by 2) the presence of inflammatory signaling molecules (i.e., interferon-γ) released by effector immune cells such as the cytotoxic T cell in the tumor microenvironment. AKT = protein kinase B; APC = antigen presenting cell; ERK = extracellular signal-regulated kinase; IFNGR = interferon-gamma receptor; IFN-γ = interferon-gamma; IRF-1 = interferon regulatory factor-1; JAK = janus tyrosine kinase; MEK = mitogen-activated protein kinase; MHC = major histocompatibility complex; NF-κB = nuclear factor kappa-light-chain-enhancer of activated B cells; PD-1 = programmed cell death protein 1; PD-L1 = programmed cell death-ligand 1; PI3K = phosphatidylinositol-3 kinase; PTEN = phosphatase and tensin homolog; STAT = signal transducer and activator of transcription; TCR = T cell receptor.

**Figure 3 cancers-13-01949-f003:**
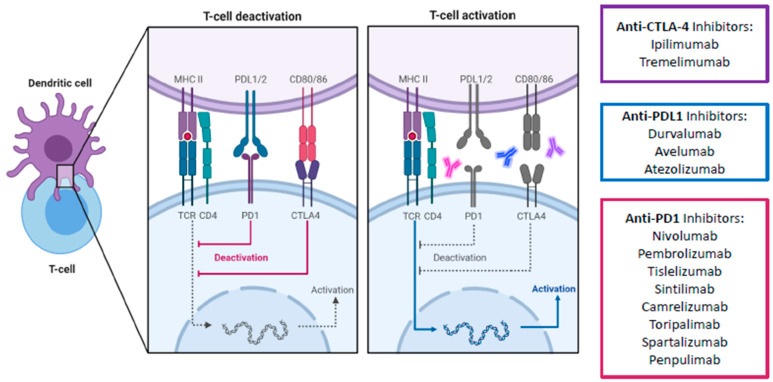
Schematic overview of main immune checkpoint inhibitors including their mechanism of action for advanced HCC treatment. The signaling cascade of the PD-1/PD-L1 and CTLA4 pathway and the corresponding therapeutical immune-checkpoint inhibitors. Due to the higher affinity of CTLA-4 to CD80/86 than the co-stimulatory CD28, CTLA-4 successfully binds to CD80/86 and antagonizes the stimulatory signal by the interaction of CD28 with CD80/86. As a result, CTLA-4 causes an inhibition of T cell activation. Similarly, the interaction of PD-L1 with its receptor PD-1 also disrupts the activation of T cells, therefore influencing immune activity in a negative way. CTLA-4, cytotoxic T-lymphocyte-associated protein 4; PD1, programmed death-1 protein; PDL1, programmed death- ligand 1; MHC II, major histocompatibility complex II; TCR, toll-cell receptor; CD4, cluster of differentiation 4. Created with biorender.com.

**Figure 4 cancers-13-01949-f004:**
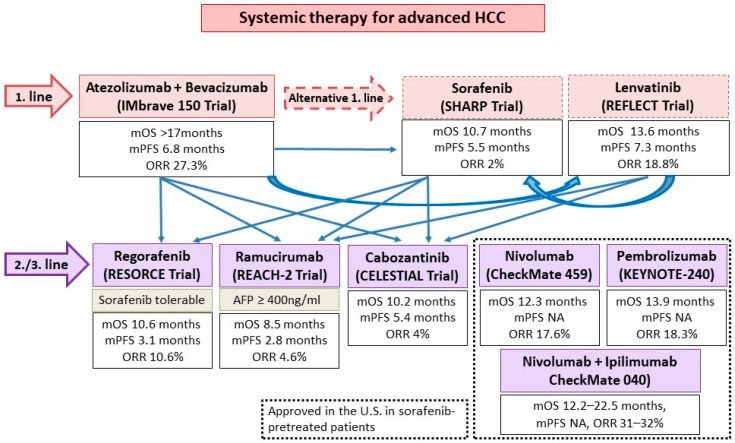
The current proposed clinical practice algorithm in terms of systemic therapies and sequencing options in advanced HCC. Adapted from [44]. NA, not available; mOS, median overall survival; mPFS, median progression free survival; ORR, overall response rate.

**Figure 5 cancers-13-01949-f005:**
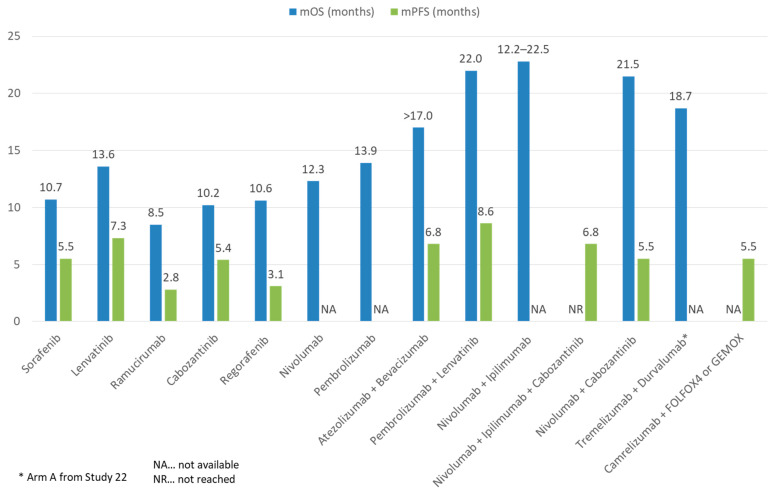
Comparison of mOS and mPFS of ICI and TKI monotherapy or combination therapy with other treatment modalities in advanced HCC.

**Figure 6 cancers-13-01949-f006:**
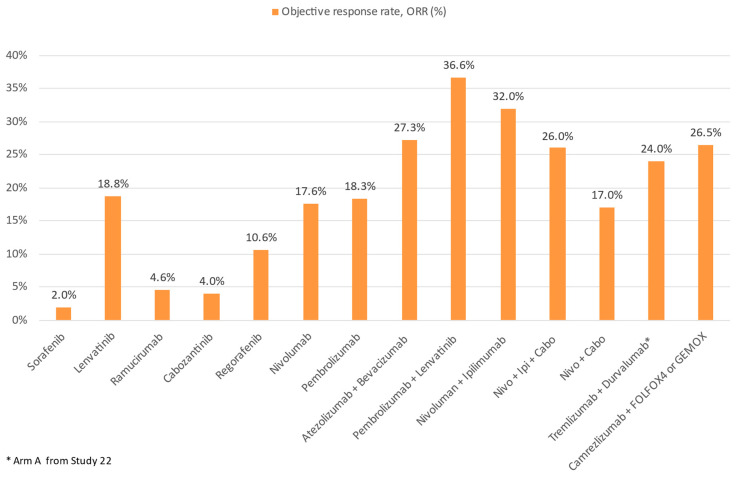
Comparison of ORR of ICI and TKI monotherapy or combination therapy with other treatment modalities in advanced HCC.

**Figure 7 cancers-13-01949-f007:**
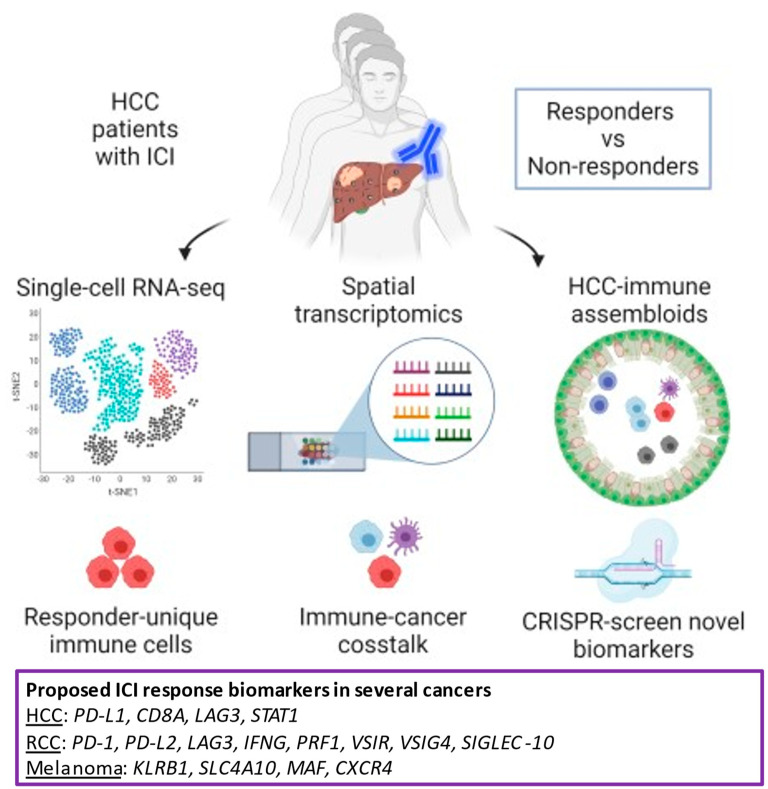
Potential biomarkers of ICI response and the future platform of translational research. Hepatocellular carcinoma (HCC) patients undergo immune checkpoint inhibitor (ICI) treatment. The patients will be divided in responders and non-responders according to clinical response. Single-cell RNA-seq of patient’s tumor samples will identify responder-unique immune cells. Spatial transcriptomics will define immune-cancer crosstalk. HCC-immune assembloids will provide experimental platforms to determine novel biomarkers such as CRISPR-based genetic screening. The high expression of inflammatory genes is proposed as potential biomarkers in HCC, renal cell carcinoma (RCC), and melanoma.

## Data Availability

The data presented in this study are openly available.

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
