# Peer review of "Recent Advances and Future Prospects in Immune Checkpoint (ICI)-Based Combination Therapy for Advanced HCC"

_cancers, 2021, doi:10.3390/cancers13081949_

Round 1

Reviewer 1 Report

Overall a good, comprehensive yet concise review

Minor suggestions in content, style and language suggested as documented in comments in PDF file

Author Response

We thank the reviewer for their comments. Style and language have been amended as suggested. In terms of content, the details of trials in question have been clarified (Lines 213-214, 276-277). The two studies which the reviewer made reference to (Line 391-399) have been added. References have been added accordingly for Lines 259-260 and Lines 547-550.

Reviewer 2 Report

This review summarize systematically the recent advances and future aspects in ICI-based combination therapy for advanced HCC. The paper is well-written. The comments I want to address

  1. Can author discuss and added real-world data into the text and Figure 5 and 6 to see the difference between trial cohort and real world difference clearly?
  2. Can authors discuss and list the exclusion criteria for each combination regimens in advanced HCC so clinicians can choose appropriate regimens safely and effectively?
  3. Can authors discuss the sequential combination and combination with "curative" treatment options?
  4. Does liquid biopsy help in choosing the combination regimen?  

Author Response

  1. Can author discuss and add real-world data into the text and Figure 5 and 6 to see the difference between trial cohort and real world difference clearly?

Response: We thank the reviewer for their comment. To the best of our knowledge, only two additional studies exist in terms of real-world efficacy of ICI combinations. One was published by our group on the combination of ICIs and cabozantinib, and the other was published recently on the use of ICI-TKI combinations to downstage HCC for subsequent liver resection. We have included these studies in our discussion (lines 320-326). In view of the heterogenous treatment regimens used in these studies, we believe they are more appropriate as part of a general discussion on ICI-TKI combinations, instead of being directly juxtaposed with other regimes in figures 5-6.

  1. Can authors discuss and list the exclusion criteria for each combination regimens in advanced HCC so clinicians can choose appropriate regimens safely and effectively?

Response: We thank the reviewer for their comment. We have added the information as suggested. In view of the large number of combinations discussed, we have included the exclusion criteria in a separate table (Table S1).

  1. Can authors discuss the sequential combination and combination with "curative" treatment options?

Response: We thank the reviewer for their comment. As mentioned in our response to comment 1, to the best of our knowledge there’s one series of patients who received ICI-TKI combinations and then received curative OT. We have added discussion on this study as mentioned (lines 322-326).

  1. Does liquid biopsy help in choosing the combination regimen?  

Response: We thank the reviewer for their comment. As mentioned in the Precision medicine of ICI section, no established biomarkers exist to predict responses to ICIs in HCC so far. Liquid biopsy may indeed have a role in the future, although further studies are required.